# Effects of Addition of Tea Polyphenol Palmitate and Process Parameters on the Preparation of High-Purity EPA Ethyl Ester

**DOI:** 10.3390/foods12050975

**Published:** 2023-02-25

**Authors:** Xuyang Ding, Fujun Liu, Rui Zheng, Xuechen Pei, Ziye Wang, Dayong Zhou, Fawen Yin

**Affiliations:** 1School of Food Science and Technology, Dalian Polytechnic University, Dalian 116034, China; 2Liao Fishing Group Limited Company, Dalian 116000, China; 3National Engineering Research Center of Seafood, Dalian 116034, China; 4Collaborative Innovation Center of Seafood Deep Processing, Dalian 116034, China

**Keywords:** EPA, high purity, ethyl esterification, urea complexation, molecular distillation, column separation

## Abstract

High-purity eicosapentaenoic acid (EPA) ethyl ester (EPA-EE) can be produced from an integrated technique consisting of saponification, ethyl esterification, urea complexation, molecular distillation and column separation. In order to improve the purity and inhibit oxidation, tea polyphenol palmitate (TPP) was added before the procedure of ethyl esterification. Furthermore, through the optimization of process parameters, 2:1 (mass ratio of urea to fish oil, g/g), 6 h (crystallization time) and 4:1 (mass ratio of ethyl alcohol to urea, g/g) were found to be the optimum conditions in the procedure of urea complexation. Distillate (fraction collection), 115 °C (distillation temperature) and one stage (the number of stages) were found to be the optimum conditions for the procedure of molecular distillation. With the addition of TPP and the above optimum conditions, high-purity (96.95%) EPA-EE was finally obtained after column separation.

## 1. Introduction

Long-chain omega-3 polyunsaturated fatty acids (n-3 LC-PUFAs), including eicosapentaenoic acid (EPA) and docosahexaenoic acid (DHA), are important components in healthy diets. So far, their health benefits have been widely reported, which include reducing triacylglycerols [1], anti-inflammation [2], anticancer activities [3] and alleviating Alzheimer’s disease [4].

n-3 LC-PUFAs are sold as soft gel capsules, aqueous emulsions and mixtures with vegetable oils, which are usually called omega-3 products. The commercial production of highly enriched n-3 LC-PUFAs is at present a major challenge for research. Single fractionation processes do not discriminate between different PUFAs. Therefore, several methods for preparing enriched fractions of n-3 LC-PUFAs have been developed, including controlled winterization [5], molecular distillation [6], urea complexation [7] and column separation [8]. Urea complexation is used to separate fatty acids based on the degree of unsaturation [9], molecular distillation is a method suitable for removing impurities [10] and column separation can further increase the purity of the obtained product [8].

So far, commercial omega-3 products contain a high purity of total n-3 LC-PUFAs, but not EPA or DHA. As the most abundant omega-3 fatty acid in the central nervous system, DHA is important for the health of babies and pregnant women [11]. Moreover, a moderate intake of EPA can effectively alleviate age-related cognitive impairment in older adults [12]. Due to the fact that the bioactivities of most bioactive substances (vitamins, n-3 LC-PUFAs, sterols and amino acids) are strongly connected to their purities [4], the enrichment of EPA and/or DHA from marine oils has attracted considerable attention. The produced high-purity EPA or DHA is conducive to developing nutritious foods which are suitable for different consumer groups.

Recently, the industry trend has shifted toward producing DHA via microalgae species. The proposed design is for a plant to cultivate *Schizochytrium* cells in the upstream process, and then extract and purify the algal oils in the downstream process [13]. Such DHA-rich algal oil is a triacylglycerol, similar to the form of DHA found in breast milk. Thus, high-purity DHA can be successfully collected from algal oils by urea collateralization [14]. Comparatively, the development of mature technology to produce high-purity EPA is urgently needed. In addition, EPA and DHA are easily oxidized and degraded during the production process (esterification, evaporation and distillation), and adding antioxidants may effectively inhibit the related oxidation [15]. Thus, with the addition of antioxidants, it is possible to improve product purity.

Given this, in order to prepare EPA ethyl esters (EPA-EE) from fish oils, which are mainly used to produce a high quality of free EPA, an integrated technique consisting of ethyl esterification, urea complexation, molecular distillation and column separation was applied. Furthermore, through the addition of tea polyphenol palmitate (TPP, a kind of antioxidant), as well as the optimization of process parameters, high-purity EPA ethyl ester was successfully obtained. This study provides fundamental data for the commercial production of high-purity EPA from fish oils.

## 2. Materials and Methods

### 2.1. Materials and Chemicals

Fish oil without added antioxidants was purchased from Qingdao Seawit Life Science Co, Ltd. (Qingdao, China). Food-grade tea polyphenol palmitate (TPP) was purchased from Guangzhou Shengtong Trading Co., Ltd. (Guangzhou, China). Urea was purchased from Shanghai Macklin Biochemical Co, Ltd. (Shanghai, China). BF_3_-methanol (14%, *w/w*, g/g) was purchased from Aladdin Bio-Chem Technology Co., Ltd. (Shanghai, China). Gas chromatography (GC)-grade n-hexane and methanol were purchased from Shanghai Macklin Biochemical Co, Ltd. (Shanghai, China). Sodium hydroxide (NaOH), ethyl ethanol (C_2_H_5_OH), sulfuric acid (H_2_SO_4_) and hydrochloric acid (HCl) were purchased from Damao Chemical Reagent Co, Ltd. (Tianjin, China).

### 2.2. Effects of TPP Added during the Ethyl Esterification Process

According to our previous studies, TPP exerts the strongest antioxidant effect among various antioxidants, including bamboo leaves, rosemary extract, vitamin E, ascorbyl palmitate, TPP, vitamin C and tea polyphenol [16,17]. Therefore, TPP was selected to be added during the ethyl esterification process.

#### 2.2.1. The Preparation of Samples Supplemented with TPP before the Ethyl Esterification Process

A mixture of fish oil (50.0 g) and 30% (*m/v*, g/mL) NaOH-H_2_O solution (150 mL) was refluxed for 2 h in an 80 °C water bath. Then, 3 mM hydrochloric acid was added to adjust the pH value to 1. Subsequently, 30 mL of water was added to above mixture, which was further extracted 3–5 times with 100–200 mL of n-hexane until the n-hexane layer was colorless. The n-hexane extract liquor was evaporated by rotary vacuum evaporation at 35 °C. Then, 20.0 g of evaporated residue was weighed, and 12 mg of TPP was added at its maximum allowable amount (600 mg/kg) permitted by the Chinese Standard GB 2760-2014 [18]. To the mixture was added 18.0 g of H_2_SO_4_-ethyl ethanol (2%, *v/v*, mL/mL) and it was placed into an 80 °C water bath. After 2.5 h, 50 mL of water was added, and the mixture was extracted 3–5 times with 100–200 mL of n-hexane until the n-hexane layer was colorless. Finally, after rotary vacuum evaporation at 35 °C, an ethyl-esterified fish oil sample supplemented with TPP before the ethyl esterification process was obtained.

#### 2.2.2. The Preparation of Samples Supplemented with TPP after the Ethyl Esterification Process

TPP was added directly to the product of ethyl-esterified fish oil at its maximum allowable amount (600 mg/kg) permitted by the Chinese Standard GB 2760-2014 [18]. Thus, an ethyl-esterified fish oil sample supplemented with TPP after the ethyl esterification process was obtained.

#### 2.2.3. Accelerated Storage Experiment

The ethyl-esterified fish oil samples supplemented with TPP before or after the ethyl esterification process were placed in an air oven at 60 °C. These samples were taken at regular intervals of 2 days until 6 days.

#### 2.2.4. Peroxide Value

According to the Chinese Standard GB 5009.227-2016 [19], the peroxide value (POV) was detected as follows: 2.0 g of ethyl-esterified fish oil was accurately weighed and dissolved in 30 mL of chloroform–acetic acid (2:3, *v/v*, mL/mL) mixed solution. Subsequently, 1 mL of starch indicator (1%, *m/v*, g/mL) and 1 mL of saturated potassium iodide solution were added. After shaking for 30 s, the mixture was placed in the dark for 3 min. Then, 100 mL of deionized water was added, which was further titrated by using 2 mM sodium thiosulfate (Na_2_S_2_O_3_) standard solution until it became colorless. POV (meq/kg) was calculated by the following formula:POV = (V − V_0_) × c × 0.1269 × 100/m
where V (mL) and V_0_ (mL) are the volumes of Na_2_S_2_O_3_ standard solution consumed in the titrating ethyl-esterified fish oil sample and the blank reagent, respectively. c (mM) is the concentration of Na_2_S_2_O_3_ standard solution. The value 0.1269 is equivalent to 1.00 mL Na_2_S_2_O_3_ standard titration solution [C(Na_2_S_2_O_3_) = 1.000 M]. m is the quantity (g) of fish oil and 100 is the conversion factor.

#### 2.2.5. Thiobarbituric Acid Reactive Substances

According to the methods reported by John et al. [20] and Wang et al. [17], the thiobarbituric acid reactive substances (TBARS) of ethyl-esterified fish oil samples were detected with slight modifications. Briefly, 0.1 g of ethyl-esterified fish oil was mixed with 2.5 mL of mixed liquor (196 mL of distilled water, 4.17 mL of concentrated hydrochloric acid solution, 0.75 g of thiobarbituric acid and 30 g of trichloroacetic acid), which was then heated for 10 min in boiling water. After cooling to room temperature, the mixture was centrifuged at 3000× *g* for 10 min. The obtained upper layer liquid was measured at 532 nm. The malondialdehyde concentration was converted to TBARS (ppm) by the following formula:TBARS = A_532_ × 2.77
where A_532_ is the absorbance value of the reaction mixture at 532 nm, and 2.77 is the conversion coefficient of malondialdehyde concentration into TBARS.

#### 2.2.6. EPA Content and DHA Content

According to the Chinese Standard GB 5009.168-2016 [19], the EPA content and the DHA content were detected as follows: 5 mg of fish oil was mixed with 200 µL of 1 mg/mL triundecylin dissolved in chloroform. Then, 2 mL of 0.5 M NaOH-CH_3_OH was added. The mixture was refluxed for 5 min in an 80 °C water bath. The reaction was carried out by adding 2 mL of BF3-CH_3_OH (14%, *w/w*, *v/v*). After 2 min, the reaction solution was extracted with n-hexane (1.5 mL). Before gas chromatography (GC) analysis, n-hexane containing fatty acid methyl ester (FAME) was filtrated through a 0.22 µm filter.

The specific parameters of GC analysis were as follows [21]: the initial temperature was 100 °C (13 min) and raised (10 °C/min) to 180 °C (6 min); then, the temperature was raised (1 °C/min) to 215 °C (20 min); finally, the temperature was raised (5 °C/min) up to 230 °C (12 min). The carrier gas comprised nitrogen, hydrogen and air. The inlet temperature and the temperature of the detector were 270 °C and 280 °C, respectively. Shunt injection was adopted, with a shunt ratio of 5:1, and the injection volume was 1 µL. The EPA or DHA content C_0_ (mg/g) in fish oil was calculated by the following formula:C_0_ = [F_i_ × A_i_ × ρ_c11_ × V_c11_ × 1.0067/(A_C11_ × m)] × F_FAMEi-FA_ × 10
where F_i_ is the influence factor of fatty acid methyl ester; A_i_ is the peak area of fatty acid methyl ester in the sample; A_C11_ is the peak area of triundecylin in the sample; ρ_c11_ is the concentration of glycerol triundecylin solution (mg/mL); V_c11_ is the volume of triundecylin added (mL); 1.0067 is the conversion coefficient of triundecylin to methyl triundecylin; m is the quantity of raw fish oil (mg); 10 is the coefficient of converting the content into the content in the sample per g; F_FAMEi-FA_ is the conversion coefficient of fatty acid methyl ester to fatty acid.

#### 2.2.7. EPA-EE Content and DHA-EE Content

Standard solutions of EPA-EE and DHA-EE with different concentrations were prepared with n-hexane as the solvent, which were then filtered by 0.22 µm filter membranes. The detection method was the same as that described in Section 2.2.6. The standard curves (x: concentration, mg/mL; y: peak area) of EPA-EE and DHA-EE were plotted as y = 725.55x + 4.33 (R^2^ = 0.9992) and y = 507.95x + 6.65 (R^2^ = 0.9997), respectively.

A certain quantity of ethyl fish oil product was mixed with 3 mL of n-hexane. The detection method was the same as that described in Section 2.2.6. The formula for calculating the EPA-EE (or DHA-EE) content C_1_ (mg/g) in the product was as follows:C_1_ = 3 × x/m
where m is the quantity of ethyl-esterified fish oil product (g), x is the concentration of EPA-EE (or DHA-EE) in the sample (mg/mL) and 3 is the n-hexane volume (mL).

#### 2.2.8. Esterification Efficiency

In the process of esterification, the esterification efficiency η (%) of EPA or DHA was calculated by the following formula:η = [(C_1_ × m_1_)/M_1_]/[(C_0_ × m_0_)/M_0_] × 100%
where m_1_ is the quantity of ethyl fish oil products (g); C_1_ is the content of EPA-EE (or DHA-EE) in ethyl-esterified fish oil products (mg/g); M_1_ is the molar mass (mg/mmol) of EPA-EE (or DHA-EE); m_0_ is the quantity of raw fish oil (g); C_0_ is the content of EPA or DHA in raw fish oil (mg/g); M_0_ is the molar mass (mg/mmol) of EPA (or DHA).

### 2.3. Parameter Optimization of Urea Complexation

According to the procedure described by Zheng et al. [22], urea complexation was performed with a slight modification. Briefly, 2 g of ethyl-esterified fish oil and urea were dissolved in 95% ethyl alcohol. The mixture was placed into a 60 °C water bath and stirred continuously until it formed a uniform solution. Mass ratios of urea to oil (g/g), including 0.5:1, 1:1, 1.5:1, 2:1 and 2.5:1, as well as mass ratios of ethyl alcohol to urea (g/g), including 2:1, 3:1, 4:1, 5:1 and 6:1, were changed by using different amounts of urea (or 95% aqueous ethyl alcohol). Subsequently, the urea inclusion compounds were allowed to crystallize at 4 °C for different times, including 2, 4, 6, 8 and 10 h. Nonurea complexation fraction (NCF, liquid phase) was obtained by filtration on a Buchner funnel under suction, and the ethyl alcohol in NCF was removed by rotary vacuum evaporation at 35 °C. Then, an appropriate amount of water was added to remove the urea residue. Finally, the upper layer was extracted with n-hexane. N-hexane in the extract liquor was removed by rotary vacuum evaporation at 35 °C. Thus, based on the EPA-EE content and the DHA-EE content of the product, the appropriate parameters were obtained, including the mass ratio of urea to oil, the mass ratio of ethyl alcohol to urea and the crystallization time.

### 2.4. Parameter Optimization of Molecular Distillation

Molecular distillation was performed in a KDL2 distiller (UIC GmbH, Hannover, Germany). The instrument contains a condensing surface of 2 dm^2^ and an evaporation surface of 5 dm^2^. The operation pressure was maintained at 0.3 mbar and the feed temperature was kept at 20 °C. The rotor speed was set at 200 rpm. The condenser temperature was set at 3 °C and the feeding flow was 1.0 ± 0.1 mL/min. The distillates and residues were obtained at different distillation temperatures (110 °C, 115 °C, 120 °C, 125 °C and 130 °C) and different stages of distillation (stage one, two or three). Thus, based on the EPA-EE content and DHA-EE content of distillates and residues, the appropriate parameters, including distillation temperature and the number of stages in distillation, were obtained.

### 2.5. Column Separation

The column separation was used for the further purification of EPA-EE. The separation was performed on a C18 column (4.6 × 250 mm, 15 mm, R2021040801) by isocratic elution with 92% (*v/v*, mL/mL) methanol–water as the mobile phase. The flow rate was 0.3 mL/min. The wavelength of detection was 220–270 nm. The injection volume was 140 µL.EPA-EE existed in the effluents from 49 min to 61 min. The purity of EPA-EE was measured through the GC method described in Section 2.2.7.

### 2.6. Statistical Analysis

The results were calculated from parallel measurements and are given as means ± standard derivations. SPSS version 16.0 software (SPSS Inc., Chicago, IL, USA) was applied for the statistical analysis. Differences between means were evaluated by one-way ANOVA (post hoc test: SNK) or independent samples t-tests. Comparisons that yielded *p* values < 0.05 (or 0.01) were considered significant.

## 3. Results

### 3.1. Effects of TPP Added during the Ethyl Esterification Process

Due to the high levels of PUFAs, including EPA and DHA, fish oils are more likely to be oxidized [23] in the process of ethyl esterification. Such oxidation will cause the decrement of nutrients, the deterioration of flavor and the generation of potentially toxic compounds [24]. POV and TBARS have been widely used to monitor the primary and secondary oxidation product in oils [25], respectively.

As shown in Figure 1A,B, the POV and TBARS values increased significantly with the extension of storage time, which indicated that all oil samples had undergone gradual oxidation. Importantly, adding tea polyphenol palmitate (TPP) before ethyl esterification (TPP-B) was more effective than adding TPP after ethyl esterification (TPP-A). For example, after 6 days of storage, the POV (or TBARS) values of the TPP-B and TPP-A groups were 67.89 meq/kg (or 4.30 mg MDA/kg) and 73.24 meq/kg (or 4.86 mg MDA/kg), respectively. Our previous study also indicated that adding TPP into the ethanol solvent during the extraction process was more effective than adding it into Antarctic krill (*Euphausia superba*) oil after the extraction process [17], consistent with the experimental results in this section.

The EPA-EE content, DHA-EE content and esterification efficiency were also calculated, and their values gradually decreased over storage time (Figure 1C,D and Figure 2). Obviously, both EPA-EE and DHA-EE in the TPP-B groups decreased more slowly. For example, after 6 days of storage, the EPA-EE contents (or DHA-EE content) of the TPP-B and TPP-A groups were 383.91 mg/g (or 242.74 mg/g) and 366.91 mg/g (or 225.62 mg/g), respectively. Furthermore, TPP added before the ethyl esterification process could also effectively improve the esterification efficiency. For example, the ethyl esterification efficiency of EPA increased from 64.05% to 71.97%, and the corresponding efficiency of DHA increased from 76.24% to 84.88%.

It is widely known that some vegetable oils such as sesame oil and hemp seed oil are rich in natural antioxidants. Many studies have shown that these natural antioxidants are effective in protecting oils from oxidation. For example, Shen et al. found that compared with oils extracted from red and white quinoa seeds, oil extracted from black quinoa seed contained higher PUFAs [26]. PUFA contents in red, white and black quinoa seeds are significantly different. The synergistic extraction of natural antioxidant components during oil extraction can also indirectly affect the oxidative stability of PUFA in oils. By contrast, tocopherol (vitamin E) and phytosterol contents in black quinoa seed oil are significantly higher than those in white and red quinoa seed oil. Nehdi et al. reported that compared with the stripped seed oil, the non-stripped seed oil shows higher oxidative stability [27]. This is mainly due to the fact that the stripped seed oil contains a low amount of tocopherol [28]. Similarly, adding TPP into crude fish oil before ethyl esterification is more effective than adding TPP into ethyl-esterified fish oil after ethyl esterification.

### 3.2. Optimization of Urea Complexation Conditions

Urea complexation is a useful technique for removing saturated fatty acids (SFAs) and monounsaturated fatty acids (MUFAs) [29]. Briefly, the oils obtained from ethyl esterification are mixed with urea–ethanol solution. SFAs and MUFAs are easily combined with urea to form a complex. After low-temperature crystallization at 4 °C, the complexes of SFAs–urea and MUFAs–urea can be removed by filtration [30]. The liquid (non-urea-complexed fraction) is enriched with polyunsaturated fatty acids (PUFAs) [31]. In the process of urea complexation, both higher and lower urea/oil ratios will lead to lower PUFA content in the product. Briefly, the complexation of urea to SFAs and MUFAs is uncompleted at a lower urea/oil ratio [32]. Meanwhile, SFAs and MUFAs are fully complexed at a higher urea/oil ratio. Unfortunately, parts of PUFAs are also inevitably “hidden” in the cavity of urea complex [33]. Crystallization time is another important factor affecting complexation efficiency. It is easily understood that the complexation efficiency of SFAs and MUFAs is too low after a short crystallization time. However, the extension of crystallization time will lead to an increase in impurities (urea) and stronger degradation of EPA-EE in the products [34]. Finally, the ethyl alcohol/urea ratio is also a vital factor affecting complexation efficiency in the process of urea complexation. Briefly, urea cannot dissolve sufficiently in ethyl alcohol at a lower ethyl alcohol/urea ratio [35]. However, at a higher ethyl alcohol/urea ratio, lower complexation efficiency will be caused by the dissolution of SFAs and MUFAs in ethyl alcohol [36].

The effect of the above three parameters on the EPA-EE content of ethyl-esterified fish oils is shown in Figure 3. Apparently, the EPA-EE content initially increased and then decreased as the values of each parameter increased. Under the conditions of 2:1 mass ratio of urea to fish oil (g/g), 6 h crystallization time and 4:1 mass ratio of ethyl alcohol to urea (g/g), the maximal EPA-EE content of 543.08 mg/g was obtained. Similar results have also been reported by other researchers. For example, Tang et al. found that the DHA-EE content of microalgae oils after urea complexation first increased, exhibited the maximum content of 604 mg/g at a 2:1 urea/oil ratio (g/g) and then decreased with the increment in this ratio [34]. Similarly, Zhai et al. reported that the maximal EPA-EE content (802 mg/g) of fish oil was obtained at a 4:1 ethyl alcohol/urea ratio (g/g) [37]. Huang et al. reported that the content of ethyl linoleic acid of bran oil was first increased and then decreased as the ratio increased from 5:1 to 15:1 (95% ethyl alcohol/urea ratio, g/g), and the maximal ethyl linoleic acid content (379.9 mg/g) was obtained [38].

### 3.3. Optimization of Molecular Distillation Conditions

#### 3.3.1. Effect of Temperature on the Efficiency of Molecular Distillation

Molecular distillation is one of the most effective purification methods to separate substances according to the difference in vapor pressure [6]. In particular, the combination of the urea complexation and molecular distillation methods can result in a significant improvement in the total concentration of PUFA (EPA and DHA) ethyl esters of marine oils [39]. Compared with other parameters, the temperature is widely regarded as the most influential variable in the process of molecular distillation [40].

In order to further improve the content of EPA-EE, as well as reduce the content of DHA-EE in the products, the fish oils obtained from urea complexation were subjected to molecular distillation. As shown in Figure 4, the effect of temperature on the EPA-EE content and the DHA-EE content in distillates and residues after one-stage distillation was evaluated. It was obvious that compared with residues, distillates contained more EPA-EE and less DHA-EE. Particularly, the contents of EPA-EE in distillates were significantly higher than those of DHA-EE. Similarly, Zheng et al. found that the content of EPA-EE in residue after molecular distillation decreased from 254.4 mg/g to 173.7 mg/g, indicating its easy transportation to the distillate [41].

Generally, the residues were collected in industrial production and some related research. For example, Fang et al. reported that after molecular distillation and urea complexation, the content of total PUFA increased from 307.2 mg/g (in crude fish oil) to 834.2 mg/g (in residue) [39]. Wen et al. reported that the content of total PUFA increased from 635 mg/g (in crude fish oil) to 789 mg/g (in residue) after molecular distillation [42]. These findings were not consistent with our results, mainly due to the different requirements (high purity of total PUFA-EE or EPA-EE) for the products.

Furthermore, the results in this section clearly indicate that the EPA-EE and DHA-EE contents in distillates first increased, exhibited the maximum contents at the temperatures of 115 °C (EPA-EE) and 120 °C (DHA-EE), and then decreased with the increment in the distillation temperature. These results are consistent with other published findings. For example, Magallanes et al. reported that the content of EPA-EE in the distillate reached the maximum value (121.14 mg/g) at 120 °C, and then decreased (e.g., 99.74 mg/g at 140 °C) with the increase in distillation temperature [43]. Similarly, Liang et al. reported that the EPA-EE content in distillate first increased, exhibited the maximum contents (155 mg/g) at the temperature of 130 °C, and then decreased with the increment in the distillation temperature [44]. Based on the differences in vapor pressures, various kinds of ethyl fatty acids can be separated from marine oils [41]. Under a certain vapor pressure, as distillation temperature increases, improvements in EPA-EE and DHA-EE contents are easily achieved [45]. However, ethyl fatty acids are a kind of heat-sensitive substance with poor oxidation stability. Obviously, a high temperature will promote the occurrence of an oxidation reaction [46].

#### 3.3.2. Effect of Distillation Stages on the Efficiency of Molecular Distillation

Molecular distillation is applied industrially to obtain fish oil and other products (free fatty acids, vitamin E and vitamin A) [47]. In many cases, the separation achieved in one-stage molecular distillation is incomplete and cannot meet the requirements [45], and thus, it is worth exploring multiple-stage schemes.

In order to produce high-purity EPA, the distillate obtained from molecular distillation also requires subsequent chromatographic column separation. Obviously, such distillate should contain more EPA-EE and less DHA-EE. As shown in Figure 5, under the distillation temperature of 115 °C, the highest EPA-EE content and the lowest DHA-EE content in distillates were obtained. Moreover, compared with distillates derived from two-stage (EPA-EE, 649.47 mg/g; DHA-EE, 260.55 mg/g) and three-stage (EPA-EE, 608.81 mg/g; DHA-EE, 295.54 mg/g) distillations, the distillate derived from one-stage distillation contained more EPA-EE (667.81 mg/g) and less DHA-EE (230.56 mg/g). Other researchers reported similar results. For example, Sosa et al. reported that compared with fatty acid ethyl esters (FAEEs) (756.3 mg/g) in distillates from two-stage distillation, FAEEs (863.4 mg/g) in distillates from one-stage distillation were more abundant [48]. Zhang et al. reported that a high purity (929.8 mg/g) of PUFA-EE was obtained from *Schizochytrium limacinum* oil after one-stage molecular distillation [6].

Furthermore, it is essential to consider that two-stage (or more) molecular distillation performed with one or more types of distillation equipment are usually expensive [49]. Given this, the highest content of EPA-EE in distillates, as well as the lowest possible costs, can be easily obtained from one-stage molecular distillation at the temperature of 115 °C.

### 3.4. The Purity of EPA-EE Obtained from Column Separation

The combination of urea complexation [50] and molecular distillation [51] can effectively increase EPA-EE content, as well as decrease DHA-EE content. However, it fails to produce high-purity EPA-EE. Therefore, column separation was applied in this study to separate EPA-EE from the above mixture obtained from the combined treatments of urea complexation and molecular distillation [52]. As shown in Figure 6, high-purity (96.95%) EPA-EE was produced from the technological process consisting of ethyl esterification, urea complexation, molecular distillation and column separation.

## 4. Conclusions

In the present study, based on the analysis of EPA ethyl ester (EPA-EE) contents, the optimum conditions of 2:1 (mass ratio of urea to fish oil, g/g), 6 h (crystallization time), 4:1 (mass ratio of ethyl alcohol to urea, g/g), distillate (fraction collection), 115 °C (distillation temperature) and one stage (the number of stages) to produce high-purity EPA-EE were obtained. Furthermore, the results of peroxide value, thiobarbituric acid reactive substances and esterification efficiency clearly indicate that adding tea polyphenol palmitate (TPP) before the procedure of ethyl esterification could effectively improve the purity and inhibit the oxidation. Thus, the technique with the addition of TPP to produce high-purity EPA-EE, which consisted of saponification, ethyl esterification, urea complexation, molecular distillation and column separation, was successfully applied.

## Figures and Tables

**Figure 1 foods-12-00975-f001:**
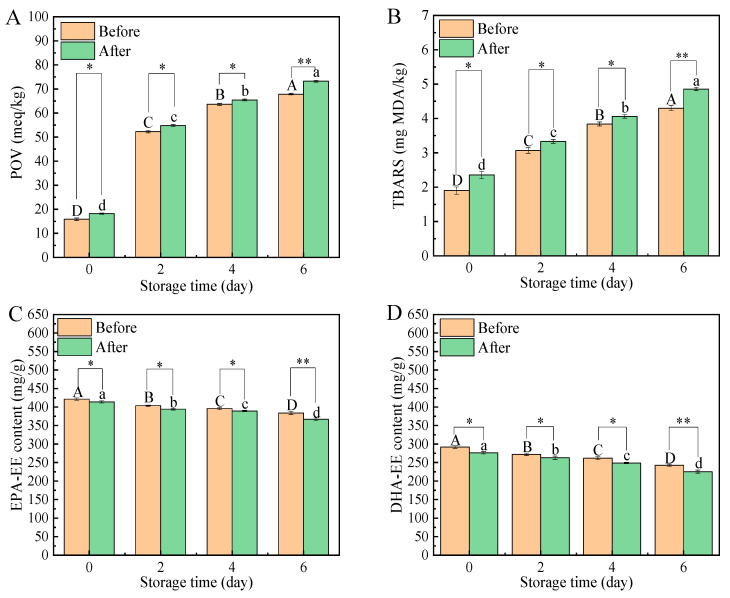
Effect of TPP added before or after the process of ethyl esterification on POV (**A**), TBARS (**B**), EPA-EE content (**C**) and DHA-EE content (**D**) of ethyl-esterified fish oils stored at 60 °C for different times. Values of different groups with different uppercase letters (A–D) or lowercase letters (a–d) are significantly different at *p* < 0.05; The asterisks (*) indicate the significant differences between the two groups with the same storage time at *p* < 0.05. The asterisks (**) indicate the significant differences between the two groups with the same storage time at *p* < 0.01.

**Figure 2 foods-12-00975-f002:**
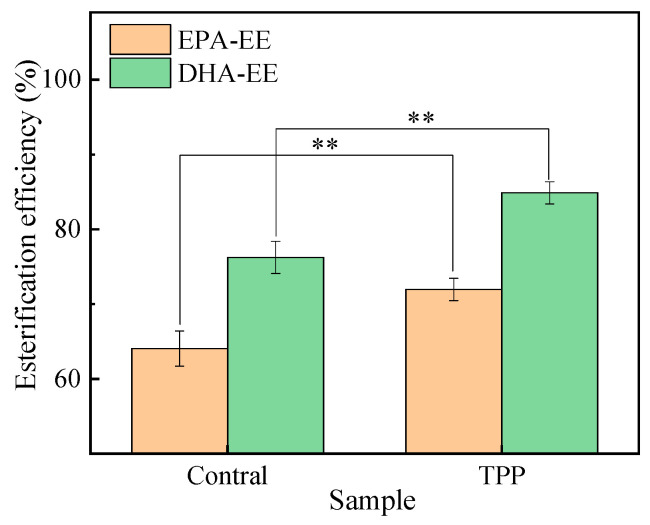
Effect of TPP on the esterification efficiency of fish oil in ethyl esterification. The asterisks (**) indicate the significant differences between the two groups supplemented with the different sample at *p* < 0.01.

**Figure 3 foods-12-00975-f003:**
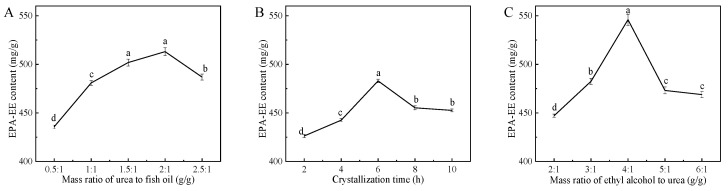
Effect of mass ratio of urea to fish oil (**A**), crystallization time (**B**) and mass ratio of ethyl alcohol to urea (**C**) on EPA-EE content of fish oils in the process of urea complexation. Values with different lowercase letters (a–d) in each panel are significantly different at *p* < 0.05.

**Figure 4 foods-12-00975-f004:**
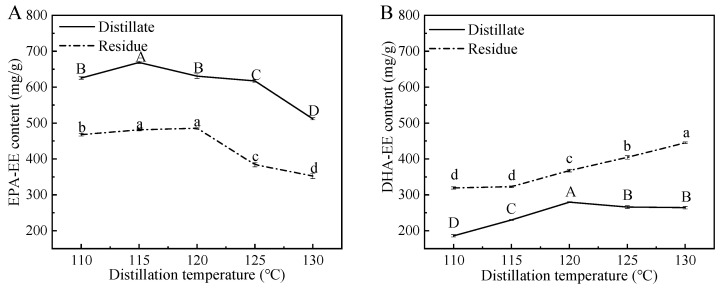
Effect of distillation temperature on EPA-EE content (**A**) and DHA-EE content (**B**) in distillates and residues after one-stage distillation. Values with different lowercase letters (a–d) or uppercase letters (A–D) are significantly different at *p* < 0.05.

**Figure 5 foods-12-00975-f005:**
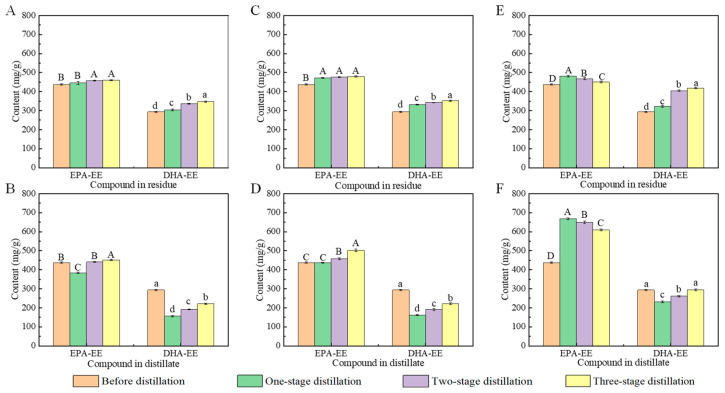
Effect of the number of stages in distillation on EPA-EE content and DHA-EE content in residues ((**A**) 85 °C; (**C**) 100 °C; (**E**) 115 °C) and distillates ((**B**) 85 °C; (**D**) 100 °C; (**F**) 115 °C). Values with different lowercase letters (a–d) or uppercase letters (A–D) are significantly different at *p* < 0.05.

**Figure 6 foods-12-00975-f006:**
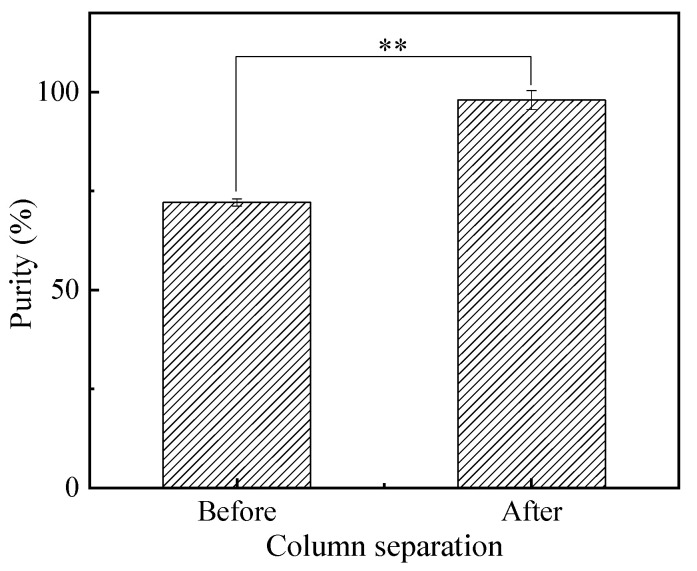
The purity of EPA-EE detected with GC in distillates after treatment with column separation. The asterisks (**) indicate the significant differences between the two groups at *p* < 0.01.

## Data Availability

Data are contained within the article.

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
