# Peer review of "Effects of Addition of Tea Polyphenol Palmitate and Process Parameters on the Preparation of High-Purity EPA Ethyl Ester"

_foods, 2023, doi:10.3390/foods12050975_

Round 1

Reviewer 1 Report

The study of adding some antioxidant component for increasing the quality of EPA-EE is interesting and the authors investigated the influence of other important operating parameters to give some  solution for  a good processing sequence. The methods used for investigation and analysis are correctly chosen and presented.

Nevertheless, the paper needs a thorough revision and reformulation to make it clear and easy to follow. Some recommendations and comments:

-          Line 100  which is in accordance with

-          Line 105  is not clear. Maybe:  samples…… were taken every 2 days…

-          Line 120  m was the quantity (g)   ( not quality!, it is measured in grams)

-          Line 123 The sentence has no predicative verb . Maybe: The method proposed by … was used…

-          Line 128  The sentence : “After cooling, centrifuge at 3000g for 10min, 200 μL of  the upper layer was measured at 532 nm”  is not clear at all.  May be:  after cooling, and centrifugation ( not centrifuge) of 3000g for 10 min, the upper layer was analysed  by measuring the absorption at 532nm …… (I assume using UV-Vis spectroscopy)

-          Line 145 temperature was raised at.. NOT temperature was heated ( or the sample was heated at)

-          Line 162: was the same with that presented in 2.2.6

-          Line 178: m was the quantity of ethyl fish oil...        ( not quality)

Author Response

Dear reviewer:

Re: foods-2224167

We are grateful to you for constructive comments and some important changes requested in our manuscript. We have accordingly carefully revised our manuscript to address all the comments made. The main revisions are explained below as “Response to Reviewer 1’s Comments”. We also invited native English-speaking collaborators to improve this manuscript. We trust that our manuscript is now ready for further processing for publication in the “Foods”.

Reviewer #1:

Comments:

  1. Line 100 which is in accordance with

Response: Thank you very much for your carefully review. According to your guideline, "Based on the additive amount allowed by Chinese Standard GB 2760-2014 [18], the TPP was added directly to the product of ethyl esterified fish oil at maximum amount is in accordance with the Chinese Standard GB 2760-2014 [18]." has been changed to "TPP was added directly to the product of ethyl esterified fish oil at its maximum allowable amount (600 mg/kg) allowed by the Chinese Standard GB 2760-2014 [18]." in the revised manuscript. Please read Page 3, Lines 97-98.

  1. Line 105 is not clear. Maybe:  samples…… were taken every 2 days…

Response: Thank you very much for your carefully review. According to your guideline, "The ethyl esterified fish oil samples added with TPP before or after the ethyl esterified process, in the accelerated storage experiment at 60 °C, take it every 2 days until 6 days." has been changed to "The ethyl esterified fish oil samples added with TPP before or after the ethyl esterified process, were placed in an air oven at 60 °C. These samples were taken at regular intervals of 2 days until 6 days." in the revised manuscript. Please read Page 3, Lines 102-104.

  1. Line 120 m was the quantity (g)   ( not quality!, it is measured in grams)

Response: Thank you very much for your carefully review. According to your guideline, "m was the quality (g) of fish oil; 100 was the conversion factor." has been changed to "m was the quantity (g) of fish oil; 100 was the conversion factor." in the revised manuscript. Please read Page 3, Lines 118-119.

  1. Line 123 The sentence has no predicative verb . Maybe: The method proposed by … was used…

Response: Thank you very much for your carefully review. According to your guideline, "Using the method by John et al. [20], according to Wang et al [17]. with a slight modification. The thiobarbituric acid reactive substances (TBARS) of ethyl esterified fish oil samples were detected as follows: 0.1 g of ethyl esterified fish oil was mixed with 2.5 mL of mixed liquor (196 mL of distilled water, 4.17 mL of concentrated hydrochloric acid solution, 0.75 g of thiobarbituric acid and 30 g of trichloroacetic acid), which was then heated for 10 min in hot water (100 °C). After cooling, centrifuge at 3000g for 10min, 200 μL of the upper layer was measured at 532 nm." has been changed to "According to the methods reported by John et al [20] and Wang et al [17], the thiobarbituric acid reactive substances (TBARS) of ethyl esterified fish oil samples were detected with slight modifications. Briefly, 0.1 g of ethyl esterified fish oil was mixed with 2.5 mL of mixed liquor (196 mL of distilled water, 4.17 mL of concentrated hydrochloric acid solution, 0.75 g of thiobarbituric acid and 30 g of trichloroacetic acid), which was then heated for 10 min in boiling water. After cooling to room temperature, the mixture was centrifugated at 3000g for 10min. The obtained upper layer liquid was measured at 532 nm." in the revised manuscript. Please read Page 3, Lines 121-128.

  1. Line 128 The sentence : “After cooling, centrifuge at 3000g for 10min, 200 μL of  the upper layer was measured at 532 nm”  is not clear at all.  May be:  after cooling, and centrifugation ( not centrifuge) of 3000g for 10 min, the upper layer was analysed  bymeasuring the absorption at 532nm …… (I assume using UV-Vis spectroscopy)

Response: Thank you very much for your carefully review. According to your guideline, "After cooling, centrifuge at 3000g for 10min, 200 μL of the upper layer was measured at 532 nm." has been changed to "After cooling to room temperature, the mixture was centrifugated at 3000g for 10min. The obtained upper layer liquid was measured at 532 nm." in the revised manuscript. Please read Page 3, Lines 126-128.

  1. Line 145 temperature was raised at.. NOT temperature was heated ( or the sample was heated at)

Response: Thank you very much for your carefully review. According to your guideline, "The specific parameters of GC analysis were as follows: the initial temperature was 100 °C for 13 min and heated at 10 °C/min to 180 °C for 6 min. Then, the temperature was heated to 215 °C at the rate of 1 °C/min for 20 min. Finally, the temperature was heated up to 230 °C at the rate of 5 °C/min and kept for 12 min." has been changed to "The specific parameters of GC analysis were as follows [21]: the initial temperature was 100 °C (13 min) and raised (10 °C/min) to 180 °C (6 min). Then, the temperature was raised (1 °C/min) to 215 °C (20 min). Finally, the temperature was raised (5 °C/min) up to 230 °C (12 min)." in the revised manuscript. Please read Page 3, Lines 141-144.

  1. Line 162: was the same with that presented in 2.2.6

Response: Thank you very much for your carefully review. According to your guideline, "The detection method was the same with 2.2.6." has been changed to "The detection method was the same with that presented in 2.2.6." in the revised manuscript. Please read Page 4, Lines 162-163.

  1. Line 178: m was the quantity of ethyl fish oil...   ( not quality)

Response: Thank you very much for your carefully review. According to your guideline, "Here, m1 was the quality of ethyl fish oil products (g);" has been changed to "Here, m1 was the quantity of ethyl fish oil products (g);" in the revised manuscript. Please read Page 4, Line 172.

Reviewer 2 Report

Manuscript number: Foods-2224167 General comment: This research group has been investigating on the effect of antioxidants on the extraction
of different oils. In this work, they extended the study to the preparation of high purity
EPA ethyl ester (EPA-EE) and the effect of TPP addition in the process.

The study was carefully planned and carried out in a muti-step procedure. This allowed assessing the effect of TPP and optimizing the following integrated steps according to the results of the EPA-EE content. The well-organized manuscript presents a comprehensive assessment of the results. The list of references allows explaining and supporting the results achieved. English language needs improving, especially in the section 2- Materials and Methods.

Some comments/questions:

·        Materials and methods-

o   2.2.1 or 2.2.2.

§  Please give information on the amount of TTP added per volume, or weight, of ethyl esterified fish oil sample.

o   2.2.3 -need to improve text in order to clearly explain the procedure.

o   2.2.6 – please confirm the extraction was carried out with hexane and not n-hexane as in other methods.

o   2.2.7-line 166. Did you mean “quality”? or “quantity?”

o   2.2.8-line 178. Did you mean “quality”? or “quantity?”. Please carry out a careful revision of the text.

o   Line 185 – Zheng et al. [22]

·        Page 10 – Figure 5 – why in chart’s vertical scale the maximum is 1000 mg/g? Using 800 mg/g as maximum, like in Figure 4, would highlight the differences between the results reached.

·        Line 366 – It is only at 115ºC that a decrease in EPA-EE content in distillate in the two and three-stage distillations is registered, as other researchers reported. Would like to read a comment from authors on the influence of temperature on this pattern.

Author Response

Dear reviewer:

Re: foods-2224167

We are grateful to you for constructive comments and some important changes requested in our manuscript. We have accordingly carefully revised our manuscript to address all the comments made. The main revisions are explained below as “Response to Reviewer 2’s Comments”. We also invited native English-speaking collaborators to improve this manuscript. We trust that our manuscript is now ready for further processing for publication in the “Foods”.

Reviewer #2:

Comments:

  1. 2.1 or 2.2.2-Please give information on the amount of TTP added per volume, or weight, of ethyl esterified fish oil sample.

Response: Thank you very much for your carefully review. According to your guideline, "Subsequently, 20.0 g of such extract product was weighed, 12 mg of tea polyphenol palmitate (TPP) was added maximum amount is in accordance with the Chinese Standard GB 2760-2014 [18]." has been changed to "Then 20.0 g of evaporated residue was weighed, and 12 mg of TPP was added at its maximum allowable amount (600 mg/kg) allowed by the Chinese Standard GB 2760-2014 [18]." in the revised manuscript. Please read Page 2, Lines 89-90.

  1. 2.3 -need to improve text in order to clearly explain the procedure.

Response: Thank you very much for your carefully review. According to your guideline, "The ethyl esterified fish oil samples added with TPP before or after the ethyl esterified process, in the accelerated storage experiment at 60 °C, take it every 2 days until 6 days." has been changed to "The ethyl esterified fish oil samples added with TPP before or after the ethyl esterified process, were placed in an air oven at 60 °C. These samples were taken at regular intervals of 2 days until 6 days." in the revised manuscript. Please read Page 3, Lines 102-104.

  1. 2.6 – please confirm the extraction was carried out with hexane and not n-hexane as in other methods.

Response: Thank you very much for your carefully review. According to your guideline, "After 2 min, the reaction solution was extracted with n-hexane (1.5 mL)." has been changed to "Subsequently, the reaction substance is extracted with hexane (1.5 mL)." in the revised manuscript. Please read Page 3, Lines 138-139.

  1. 2.7-line 166. Did you mean “quality”? or “quantity?”

Response: Thank you very much for your carefully review. According to your guideline, "A certain quality of ethyl fish oil product was accurately weighed, dissolved in 3 mL n-hexane (GC grade), filtered by 0.22 µm filter membrane for detection." has been changed to "A certain quantity of ethyl fish oil product was mixed with 3 mL of n-hexane." in the revised manuscript. Please read Page 4, Line 162.

  1. 2.8-line 178. Did you mean “quality”? or “quantity?”. Please carry out a careful revision of the text.

Response: Thank you very much for your carefully review. According to your guideline, "Here, m1 was the quality of ethyl fish oil products (g);" has been changed to "Here, m1 was the quantity of ethyl fish oil products (g);" in the revised manuscript. Please read Page 4, Lines 172.

  1. Line 185 – Zheng et al. [22]

Response: Thank you very much for your carefully review. According to your guideline, "The urea complexation was performed according to the procedure described by Zhang et al. [22] with a slight modification." has been changed to "According to the procedure described by Zheng et al. [22], the urea complexation was performed with a slight modification." in the revised manuscript. Please read Page 4, Lines 178-179.

  1. Page 10 – Figure 5 – why in chart’s vertical scale the maximum is 1000 mg/g? Using 800 mg/g as maximum, like in Figure 4, would highlight the differences between the results reached.

Response: Thank you very much for your carefully review. According to your guideline, Figure 5 has already been redrawn in the revised manuscript, which is shown as follows:

Figure. 5. Effect of the number of stages in distillation on EPA-EE content and DHA-EE content in residues (A-85°C; C-100°C; E-115°C) and distillates (B-85°C; D-100°C; F-115°C). Values with different lower case letters (a–d) or upper case letters (A–D) are significantly different at p < 0.05.

  1. Line 366 – It is only at 115ºC that a decrease in EPA-EE content in distillate in the two and three-stage distillations is registered, as other researchers reported. Would like to read a comment from authors on the influence of temperature on this pattern.

Response: Thank you very much for your carefully review. The related discussions are shown in 3.3.1. Please read Page 9, Lines 333-347. The screenshot is shown as follows:

Reviewer 3 Report

The paper is very good, but you must express all figures to tables 

Express all results in tables

Author Response

Dear reviewer:

Re: foods-2224167

We are grateful to you for constructive comments and some important changes requested in our manuscript. We have accordingly carefully revised our manuscript to address all the comments made. The main revisions are explained below as “Response to Reviewer 3’s Comments”. We also invited native English-speaking collaborators to improve this manuscript. We trust that our manuscript is now ready for further processing for publication in the “Foods”.

Reviewer #3:

Comments:

  1. The paper is very good, but you must express all figures to tables

Express all results in tables

Response: Thank you very much for your carefully review. As you said, we tabulated the results of our experiments in the early stages of writing this article. However, we found that this is not conducive to reading. Instead, by graphing the results of the experiment, the reader can intuitively see the difference between the results of different groups. To be honest, bar charts are better. If you still insist on changing the figures to the tables, we will make the corresponding changes in the next revision.

Reviewer 4 Report

Thank you for the opportunity to review this article.

The authors evaluated the Effects of addition of tea polyphenol palmitate and process pa- 3 rameters on the preparation of high purity EPA ethyl ester.

The manuscript is generally well written in good order.

Some minor remarks follow.

Line 13. Please give the whole name af EPA

Please among results develop the discussion more.

Please write all the references with the same way and according journal's guidelines.

Please remove the very old references.

Author Response

Dear reviewer:

Re: foods-2224167

We are grateful to you for constructive comments and some important changes requested in our manuscript. We have accordingly carefully revised our manuscript to address all the comments made. The main revisions are explained below as “Response to Reviewer 4’s Comments”. We also invited native English-speaking collaborators to improve this manuscript. We trust that our manuscript is now ready for further processing for publication in the “Foods”.

Reviewer #4:

Comments:

  1. Line 13. Please give the whole name of EPA

Response: Thank you very much for your carefully review. According to your guideline, "High purity EPA ethyl ester (EPA-EE) could be produced from the integrated technique consisted of saponification, ethyl esterification, urea complexation, molecular distillation and column separation." has been changed to "High purity eicosapentaenoic acid (EPA) ethyl ester (EPA-EE) could be produced from the integrated technique consisted of saponification, ethyl esterification, urea complexation, molecular distillation and column separation." in the revised manuscript. Please read Page 1, Lines 13-15.

  1. Please among results develop the discussion more.

Response: Thank you very much for your carefully review. The related discussions have been added in the revised manuscript. The relevant screenshots are as follows:

  1. Please write all the references with the same way and according journal's guidelines.

Response: Thank you very much for your carefully review. The reference section has been carefully revised. The relevant screenshots are as follows:

  1. Please remove the very old references.

Response: Thank you very much for your carefully review. According to your guideline, "Michel, L. Editorlal: health benefits of docosahexaenoic acid (DHA). Pharmacological Research, 1999, 40(3): phrs.1999.0497." has been changed to " Myhre, P. L.; Kalstad, A. A.; Tveit, S. H.; Laake, K.; Schmidt, E. B.; Smith, P.; Nilsen, D. W. T.; Tveit, A.; Solheim, S.; Arnesen, H. & Seljeflot, I. Changes in eicosapentaenoic acid and docosahexaenoic acid and risk of cardiovascular events and atrial fibrillation: A secondary analysis of the OMEMI trial. Journal of Internal Medicine, 2022, 291(5), 637–647."; "Wanasundara, U. N. & Shahidi F. Concentration of omega 3-polyunsaturated fatty acids of seal blubber oil by urea complexation: optimization of reaction conditions. Food Chemistry, 1999, 65(1), 41-49." has been changed to "Drenjancevic, I. & Pitha, J. Omega-3 polyunsaturated fatty acids-vascular and cardiac effects on the cellular and molecular level (narrative review). International Journal of Molecular Sciences, 2022, 23(4), 2104."; "Wanasundara, U. N. & Shahidi, F. Concentration of omega 3-polyunsaturated fatty acids of seal blubber oil by urea complexation: optimization of reaction conditions. Food Chemistry, 1999, 65(1), 41-49." has been changed to "Zhang, W. W.; Ge, S. S.; Li, K.; Li, K.; Xu, J.; Gan, J. & Zhang, H. Enrichment of polyunsaturated fatty acids from Phyllanthus emblica L. seed oil by urea inclusion. Current Topics in Nutraceutical Research, 2019, 17(4), 406–424." in the revised manuscript. Please read Page 11, Lines 415-417, 420-421; Page 12, Lines 475-476.
